



# Maximum Entropy Distribution of Rainfall Intensity and Duration – MEDRID: a method for precipitation temporal downscaling for sediment delivery assessment.

Pedro Henrique Lima Alencar[1,2], Eva Nora Paton[1], and José Carlos de Araújo[2]

[1]Technische Universität Berlin, Institut für Ökologie, Ernst-Reuter-Platz 1 10587 Berlin, Germany
[2]Universidade Federal do Ceará, Departamento de Engenharia Agrícola, Campus do Pici Fortaleza, Brazil

**Correspondence:** Pedro Alencar (pedro.alencar@campus.tu-berlin.de)

**Abstract.** Scarcity of precipitation data is yet a problem in erosion modelling, especially when working in remote and data scarce areas. While much effort was made to use remote sensing and reanalysis data, they are still considered to be not completely reliable, notably for sub-daily measures such as duration and intensity. A way forward are statistical analyses, which can help modellers to obtain sub-daily precipitation characteristics by using daily totals. In this paper, we propose

a novel method (Maximum Entropy Distribution of Rainfall Intensity and Duration - MEDRID) to assess the duration and intensity of sub-daily rainfalls relevant for modelling of sediment delivery ratios. We use the generated data to improve the sediment yield assessment in seven catchments with area varying from $10^{-3}$ to $10^{+2}$ $km^2$ and broad timespan of monitoring (1 to 81 years). The best probability density function derived from MEDRID to reproduce sub-daily duration is the generalised gamma distribution (NSE = 0.98), whereas for the rain intensity it is the uniform (NSE = 0.87). The MEDRID method coupled

with the SYPoME model (Sediment Yield using the Principle of Maximum Entropy) represents a significant improvement over empirically-based SDR models, given its average absolute error of 21% and a Nash-Sutcliffe Efficiency of 0.96 (rather than 105% and -4.49, respectively).

## 1 Introduction

Climate change challenges our capacity to preserve natural resources, such as clean water and productive soil. The Food

and Agriculture Organisation named erosion as one of the most relevant threats to soil conservation and agriculture (FAO, 2019). Climate change is blamed for erosion rates increasing by nearly 17% in the USA and Europe until 2050 due to higher rainfall erosivity (Nearing, 2004; Panagos et al., 2017). This is why soil erosion turned into a key challenge for the Sustainable Development Goals of the UN (Keesstra et al., 2016; Borrelli et al., 2017). Soil erosion also imposes a threat to water supply, as pollutants and heavy metals are transported along with sediment, augmenting toxicity, turbidity and eutrophication in aquatic

environments (Coelho et al., 2017; Li et al., 2020).

In addition, 30% of all land on Earth has arid or semiarid climate (Sivakumar et al., 2005), which causes places to be especially vulnerable to climate change and soil erosion (Huang et al., 2015). Special attention is required for semiarid regions, since



they house and sustain over 14% of the global population and around 70% of the dry-land population (Huang et al., 2015).
Arid and semiarid areas are commonly affected by data scarcity, particularly in Africa, Asia and South America (Sanyal et al.,
2014; Worqlul, 2017; Rezende de Souza et al., 2021). It is necessary to improve sedimentological and other models, in
order to better estimate the amount of sediment reaching water bodies. Modellers normally have information only on daily
precipitation data, yet sub-daily processes play a crucial role in sediment transport, as a substantial amount occurs during
high-intensity storms (Srinivasan and Galvão, 2003; Shrestha et al., 2019). Therefore, we need a methodology to downscale
precipitation duration and to improve erosion models at the sub-daily scale.

Diverse branches of water sciences point out the use of stochastic methods in hydrology as being the next generation of models
(Sidorchuk, 2009; Singh, 2018). In this context, a powerful tool deployed in several studies over the last decades is the Principle
of Maximum Entropy (PoME – Shannon, 1948; Jaynes, 1957). The first applications of PoME in water sciences were proposed
by Chiu (1987) and by Singh and Chowdhury (1985) for modelling velocity distribution in open channels. Since then, several
other applications in hydrology, hydraulics and sedimentology have been presented (Sterling and Knight, 2002; Singh, 2011;
Chen et al., 2017; Kumbhakar et al., 2020).

de Araújo (2007) proposed a PoME-based model to assess sediment yield and reservoir siltation. The model (Sediment Yield
using the Principle of Maximum Entropy – SYPoME), however, requires sub-daily data, such as rainfall duration and intensity
measurements, which is often unavailable in arid and semiarid regions (Pilgrim et al., 1988), as the Brazilian northeast region.
According to the Brazilian Water Management Agency (ANA, 2019), the country's semiarid region has 2,163 operating rainfall
stations connected to the national weather monitoring system, which averages of one rain gauge per 462 km$^2$. Most of those
instruments are standard Ville de Paris gauges, providing only daily precipitation. Only 36 are active and reliable automatic
stations providing sub-daily precipitation data - one every 27,800 km$^2$, on average (Figure S1 - Supplementary material). The
gauging station density is much lower than in other regions (e.g., the density of automatic stations is one per 3,600 km$^2$ in the
United States and 77 km$^2$ in Italy – NOAA, 2013; Baldassarre et al., 2006). The data series are also not long, only 16 stations
have more than 15 years of continuous data.

The Brazilian northeast (10$^6$ km$^2$) has an average annual temperature varying between 20 and 28ºC and is characterised by a
high temporal and spatial rainfall variability (Medeiros and de Araújo, 2014), with average annual rainfall between 400 mm
and 800 mm (increasing towards the coast – Cadier, 1994; Andrade et al., 2020) and evapotranspiration between 2000 and
2600 mm per year (de Figueiredo et al., 2016). The vegetation is mainly Caatinga, formed by deciduous broadleaf bushes. The
largest part of the region is placed over Precambrian crystalline bed-rock with shallow soils. In these areas, groundwater is
scarce and usually salty (Gaiser et al., 2003; Marengo et al., 2013). The simultaneous occurrence of such geological features,
concentrated precipitation patterns and high evaporation rates leads to a scenario where rivers are predominantly intermittent
(Malveira et al., 2012; Montenegro and Ragab, 2012). As a result, water for over twenty million people living in the Brazilian
northeast region is mainly supplied by reservoirs (Mamede et al., 2012). The region has a concentration of reservoirs as high
as one per 5 km$^2$ (Peter et al., 2014). Due to excessive erosion and eutrophication, however, reservoir siltation is one of the key
threats to the water supply in the region (Coelho et al., 2017; Gil et al., 2020).





Our objectives are: (1) to propose a temporal down-scaling method to estimate sub-daily precipitation data from daily precipitation data based on the Principle of Maximum Entropy (MEDRID); (2) to assess the method quality when implemented on ungauged regions (spatial-scalability); and (3) to evaluate the effect of the method on the performance of long-term sediment
yield modelling.

In order to achieve these objectives, measured data of high-resolution precipitation were used to calibrate and validate the MEDRID method, and the statistical distance measures after Kullback (1978) and Fedotov et al. (2003) to assess spatial scalability. Measured sediment yield data of seven catchments of different size and series duration were employed to test and validate the improved sediment yield modelling using scaled precipitation together with the model by de Araújo (2007), which
is based on entropy equations and quantifies gross erosion by means of the Universal Soil Loss Equation (USLE).

## 2   Materials and Methods

Sediment yield can be quantified by multiplying gross erosion and sediment delivery ratio (SDR – Maner, 1958; Sharda and Ojasvi, 2016; Llena et al., 2021). These terms are highly nonlinear, and deterministic models do not always account for their uncertainties (Sidorchuk, 2009; Royall and Kennedy, 2016; Llena et al., 2021). Therefore, such processes need to be modelled
stochastically and event-wise (Sidorchuk, 2009; Gupta et al., 2020). In this study, sediment yield of sub-daily events was quantified using the Principle of Maximum Entropy (PoME). To incorporate sub-daily rainfall information, we developed temporal-downscaling equations to assess the effective rainfall duration ($D$) and its respective 30-minute intensity ($I_{30}$). As proposed by de Araújo (2007), the rainfall duration was drawn on to calculate the SDR, and the $I_{30}$ to calculate the erosivity factor of the Universal Soil Loss Equation (Wischmeier and Smith, 1978), so as to assess gross erosion.

A new method (Figure 1) was proposed to estimate sediment yield: it consists of an entropy-based approach to downscale rainfall duration and intensity (the MEDRID – Maximum Entropy Distribution of Rainfall Intensity and Duration method). We coupled MEDRID with the SYPoME model to determine an event-wise SDR (de Araújo, 2007).



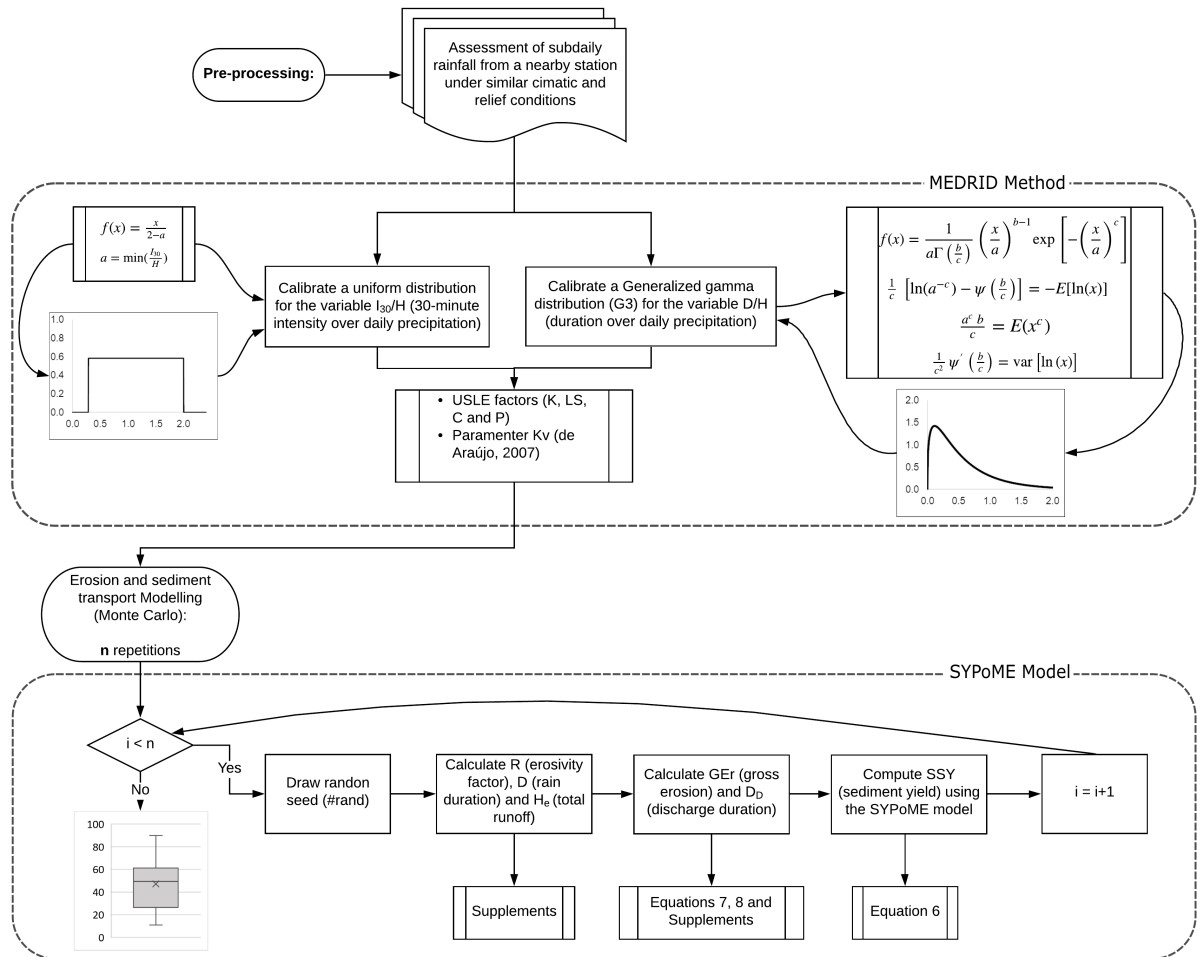

**Figure 1.** Flowchart of the proposed model. The processing is divided in two main parts, the MEDRID method and the SyPOME model. The two parts are coupled by a Monte Carlo process with multiple random seeds generated.

## 2.1 Maximum Entropy Distribution of Rainfall Intensity and Duration – MEDRID Method

Two sub-daily variables were selected to be assessed from daily rainfall data: (1) the Duration- Precipitation ratio $D/H$ ($D$ for Duration and $H$ for total daily precipitation) and (2) intensity-precipitation ratio $I_{30}/H$ (where $I_{30}$ stands for 30-minute intensity). Three probability density functions were tested to fit $D/H$ frequencies: the beta (B3), the gamma (G2) and the generalised gamma (G3) distributions (Stacy, 1962; Chen et al., 2017). For the intensity-precipitation ratio ($I_{30}/H$), two probability density functions were tested: the beta (B3) and the uniform distribution. After calibrating the equations using the Principle of Maximum Entropy (Singh, 1998), we tested the best fitting equations to measured data, as well as spatial scalability.





Table 1 presents the three Probability Density Functions (PDF – beta, gamma and generalised gamma), their constrains and the respective system of equations for parameterization. $\Psi(\cdot)$ is the digamma function, the first derivative of $\Gamma(\cdot)$, the gamma function. $\Psi'(\cdot)$ is the tri-gamma function, the second derivative of $\Gamma(\cdot)$. The terms $a$, $b$ and $c$ in the three distributions are parameters obtained maximising entropy using the Lagrange Multipliers Method (Kumbhakar et al., 2020). The systems of equations in Table 1 can be solved using empirical data (e.g. rain gauge readings, as for this study – Singh and Chowdhury, 1985). The parameter $r$ in the beta distribution (B3) is a scale factor. For this specific distribution, the random variable $X \in [0,1]$. The systems of equations were solved with help of the software Octave (v. 5.1.0.0).

Additionally, sub-daily data are scarce and stations may cover a large area. It is important to assess the loss in performance of the method when using data from a distant station. This loss of performance can be measured as the difference between the calibrated PDF for the weather station and the expected PDF, if the region of study had such a station. In this study we compared the variations among four stations with sub-daily data (Aiuaba, Sobral, Sumé and Gilbués) using the Kullback-Leibler Divergence (Kullback and Leibler, 1951) and the Kolmogorov-Smirnov Distance (Kolmogorov, 1933; Smirnov, 1939). These statistical measures allow to find similarities between the areas and, therefore, to determine, which areas can be modelled with which calibrated PDF without a significant performance loss.

Let $m$ and $n$ be two populations (sets) – in our study, automatic stations – each with an associated PDF $p_m$ and $p_n$. Kullback and Leibler (1951) present a measure that allows to compare how different those two distributions are. Known as the Kullback-Leibler Divergence, the $D_{KL}$ is an asymmetric measure, given by Equation 1.

$$D_{KL}(P_m \parallel P_n) = I(m:n) = \int\limits_{0}^{+\infty} p_m(x)\ln\left[\frac{p_m(x)}{p_n(x)}\right] dx \tag{1}$$

$$J(m,n) = I(m:n) + I(n:m) \tag{2}$$

where $p_m$ and $p_n$ are continuous probability distributions. $I(m:n)$ can be understood as the loss of information if the population $m$ is modelled using $p_n$ instead of $p_m$. Furthermore, Kullback (1978) introduces a symmetric measure, given by Equation 2. $J(m,n)$ is also a measure of divergence between the distributions $p_m$ and $p_n$ and can be interpreted as how easily we can distinguish the two distributions, henceforth called Symmetric Divergence.





**Table 1.** Parameterization of PDFs Beta (B3), Gamma (G2) and generalised Gamma (G3). We present the list of constrains used for each equation and the obtained system after solving with the Lagrange multipliers method.

| Equation | | B3 | G2 | G3 |
|---|---|---|---|---|
| PDF | | $f(x) = \dfrac{\Gamma(a+b)}{\Gamma(a)+\Gamma(b)} \left(\dfrac{x}{r}\right)^{a-1} \left(1-\dfrac{x}{r}\right)^{b-1}$ | $f(x) = \dfrac{1}{a\Gamma(b)} \left(\dfrac{x}{a}\right)^{b-1} \exp\left(-\dfrac{x}{a}\right)$ | $f(x) = \dfrac{c}{a\Gamma\left(\frac{b}{c}\right)} \left(\dfrac{x}{a}\right)^{b-1} \exp\left[-\left(\dfrac{x}{a}\right)^{c}\right]$ |
| constrains | *i.* | $\displaystyle\int_0^1 f\left(\dfrac{x}{r}\right) d\left(\dfrac{x}{r}\right) = 1$ | $\displaystyle\int_0^{+\infty} f(x)dx = 1$ | $\displaystyle\int_0^{+\infty} f(x)dx = 1$ |
| | *ii.* | $\displaystyle\int_0^1 \dfrac{x}{r} f\left(\dfrac{x}{r}\right) d\left(\dfrac{x}{r}\right) = E\left[\ln\left(\dfrac{x}{r}\right)\right]$ | $\displaystyle\int_0^{+\infty} x f(x)dx = E(x)$ | $\displaystyle\int_0^{+\infty} x^q f(x)dx = E(x^q)$ |
| | *iii.* | $\displaystyle\int_0^1 \ln\left(1-\dfrac{x}{r}\right) f\left(\dfrac{x}{r}\right) d\left(\dfrac{x}{r}\right) = E\left[\ln\left(1-\dfrac{x}{r}\right)\right]$ | $\displaystyle\int_0^{+\infty} \ln(x) f(x)dx = E[\ln(x)]$ | $\displaystyle\int_0^{+\infty} \ln(x) f(x)dx = E[\ln(x)]$ |
| system | *i.* | $E\left[\ln\left(\dfrac{x}{r}\right)\right] = \psi(a) - \psi(a+b)$ | $ab = \bar{x}$ | $\dfrac{1}{c}\left[\ln(a^{-c}) - \psi\left(\dfrac{b}{c}\right)\right] = -E[\ln(x)]$ |
| | *ii.* | $E\left[\ln\left(1-\dfrac{x}{r}\right)\right] = \psi(b) - \psi(a+b)$ | $\psi(b) - \ln(b) = E[\ln(x)] - \ln(\bar{x})$ | $\dfrac{a^c b}{c} = E(x^c)$ |
| | *iii.* | | | $\dfrac{1}{c^2}\psi'\left(\dfrac{b}{c}\right) = \text{var}[\ln(x)]$ |





The Kolmogorov-Smirnov distance ($\delta$ – Eq. 3) is the maximum distance between two distributions in their domain and is
related to the Kullbach-Leibler divergence by the Pinsker's inequaly (Eq. 4).

$$\delta(P_m, P_n) := sup \left\{ \left| \int_0^x p_m(x)dx - \int_0^x p_n(x)dx \right| \right\} \tag{3}$$

$$\delta(P_m, P_n) \leq \sqrt{\frac{1}{2} D_{KL}(P_m || P_n)} \tag{4}$$

It is also important to note that $J$ is not an actual distance, while $\delta$ is. The PDFs obtained for each of the four stations will
be compared pairwise. The lower the values of $D_{KL}$ and $\delta$ are, more alike are the two distributions and lower is the loss of
information between the areas.

### 2.1.1   Other literature approach

de Araújo (2017) also attempted to assess event duration using stochastic modelling using Equations 5. $D$ is Duration and $H$
daily precipitation. $S_\bullet$ is the standard deviation of the sample. $j$ is a counter index ($j$-th event). $\chi$ is a random number such
that $\chi^j \in [0, \chi_{max}]$. $\chi_{max}$ is calibrated for each watershed. The author proposes that for each event $j$, at least 20 values of $\chi^j$
should be drawn. The simulated duration $D$ would be the arithmetic average of the 20 produced results.

$$D^j = \bar{D} + k^j S_D \tag{5a}$$

$$k^j = \frac{H^j - \bar{H}}{S_H} \chi^j \tag{5b}$$

$$\frac{\bar{D} - D^j}{\bar{H} - H^j} = \frac{S_D}{S_H} \chi^j \tag{5c}$$

### 2.2   Sediment Yield-PoME – SYPOME Method

de Araújo (2007) proposed an entropy-based model for event-based SDR (Equation 6) and sediment yield ($SSY$ – Mg km$^{-1}$
yr$^{-1}$). $\bar{\varepsilon}$ (Mg km$^{-1}$ yr$^{-1}$) is the gross erosion obtained, for example, by using the Universal Soil Loss Equation – USLE (Wis-





chmeier and Smith, 1978), $L_0$ the hill slope length (m), $L_m$ the maximum sediment travel distance (m), $x_0$ is the initial position of erosion in the hillslope and $\lambda$ is a Lagrange multiplier.

$$SSY = \bar{\varepsilon} \times \textbf{SDR} = \bar{\varepsilon} \times \frac{e^{\lambda L_m}\left(L_0 - x_0\right)\lambda - \left(e^{\lambda(L_0 - x_0)} - 1\right)}{\lambda L_0 \left(e^{\lambda(x_0 + L_m)} - 1\right)} \tag{6}$$

The, the $\textbf{SDR}$ is the ratio of Sediment Yield ($SSY$) and mobilised sediment ($\bar{\varepsilon}$). The $SDR$ is physically constrained to a closed interval ($SDR \in [0,1)$), and it can be interpreted as the average probability of a detached particle reaching the river system (de Araújo, 2007). The SYPoME model uses as input the duration of the sub-daily precipitation which, in our case, is not known. The MEDRID method can solve this gap, based on daily precipitation.

### 2.3 Monte Carlo and MEDRID-SYPoME coupling

A Monte Carlo approach was used to adapt the SYPoME model (de Araújo, 2007) and its output to an interval of possible values of sediment yield associated to a probability function (Vrugt et al., 2008). The results were compared with measured data from seven catchments (Fig. 2 and Table 2) and values from literature model (Maner, 1958).

Using the MEDRID method we can find the probability distribution function (PDF) for the duration-precipitation ratio $\frac{D}{H}$. To model the inherent uncertainty of the duration-precipitation ratio we used the Monte Carlo approach. For each event in the time
interval $\Delta t$, a large number of random seeds ($\#_{rand} \in [0,1]$ - Eq. 7) are generated and used as input in the calibrated PDF to assess the duration (Figure 1).

$$\#_{rand} = F\left(x \leq \frac{D}{H}\right) = \int_0^{D/H} f(x)dx \tag{7}$$

where $f$ is the calibrated PDF according with Table 1 and $F$ the associated cumulative distribution function of $x$. Solving equation 7 for $D/H$, with known $H$, we can obtain the rainfall duration for each random seed $\#_{rand}$. The set of pairs $(D, H)$
is the used as input for the SYPoME model.




## 2.4 Gross erosion and siltation assessment

To estimate gross erosion in the catchments we used the Universal Soil Loss Equation (Eq. 8 – Wischmeier and Smith 1978; Bagarello et al. 2020). A more detailed description of each factor and the values for the study ares can be found in the supplements to this paper. Siltation ($\Delta V$) and Sediment Yield are proportional and related according to Equation 9.

$$\bar{\varepsilon} = R\,K\,L S\,C\,P \tag{8}$$

$$SSY = \frac{\Delta V\,\rho_s}{\eta\,A\,\Delta t} \tag{9}$$

where $\Delta V$ is the volumetric siltation, or the reservoir capacity loss (in m$^3$), $\rho_s$ is the bulk density of the silted sediment (in Mg m$^{-3}$), $\eta$ the trap efficiency of the reservoir (using, e.g., the method by Brune, 1953), $A$ is the catchment area in hectares and $\Delta t$ the interval of time in analysis.

In order to assess the performance gain by using the MEDRID+SYPoME model, we compared the measured data with empirically-based SDR equations (Sharda and Ojasvi, 2016). Gaiser et al. (2003) found that, for the Brazilian northeast region, the most fit among those equations is the one by Maner Maner (1958, hereafter Equation 10). Simplício et al. (2020) had the same result for the dry Cerrado region of Gilbués (Fig. 2).

$$\text{SDR} = \exp\left[2.943 - 0.824\log_{10}\left(\frac{F_L}{F_R}\right)\right] \tag{10}$$

$F_L$ (m) is the length factor, measured as the maximum distance in the catchment with a straight line from the outlet to the water divide approximately parallel to the main river. $F_R$ (m) is the relief factor, calculated as the difference between the outlet altitude and the average altitude of the water divide.

## 2.5 Study area

We selected seven catchments in three different states of the Brazilian northeast, all under dry conditions (Figure 2) to test the
method approach for precipitation downscaling (MEDRID) and sediment yield assessment model (SYPoME). The catchments vary widely in area and availability of data (number of years in a time series). They also vary in terms of Land Use and Land Cover. The characteristics of the studied catchments are listed in Table 2.





The Brazilian northeastern region houses the country's semiarid region (BSh climate, according to the Köpper Classification – Gaiser et al., 2003) and the Caatinga Biome. The Caatinga is the largest tropical dry forest in the world and houses the highest
endemic genera of all (Miles et al., 2006; Silva and Souza, 2018). The main economic activities in the region are agriculture (especially maize, beans and soybeans), livestock and fishing (Coelho et al., 2017). Due to deleterious practices in agriculture and overgrazing, the degraded area surpassed 72,000 km$^2$ in the Brazilian Drylands (ca. 8% its original area – Tomasella et al., 2018).

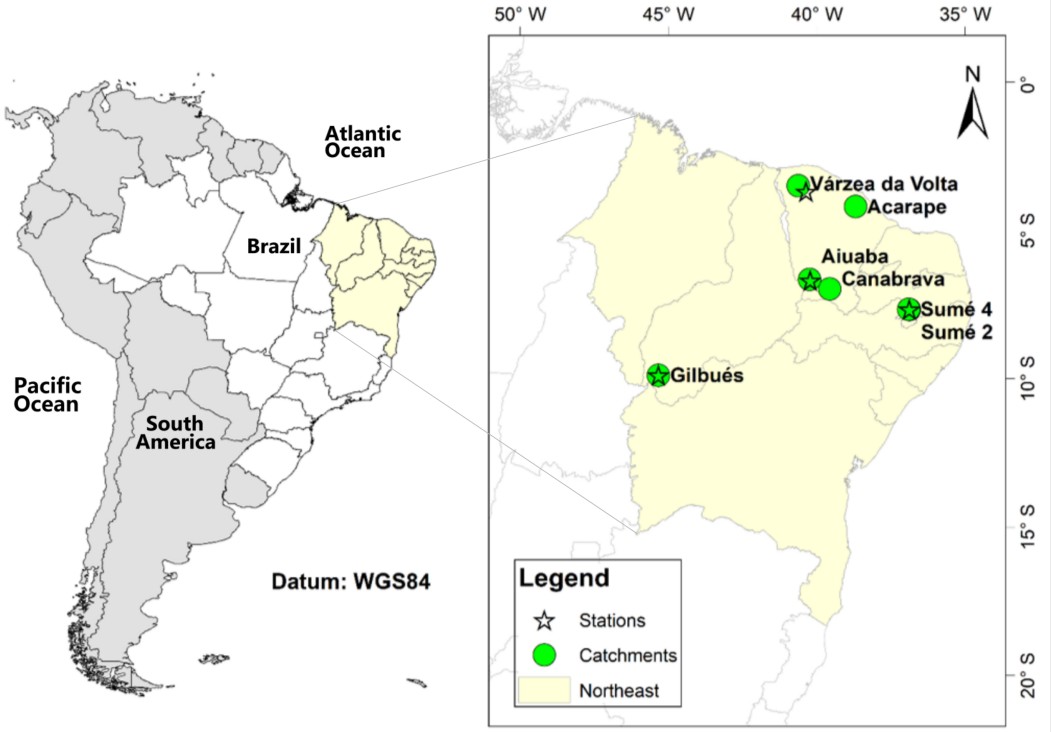

**Figure 2.** Location of study areas (catchments) and automatic rain gauges. All areas are located in the Brazilian northeast.

As presented in Section 1, the Brazilian northeast region suffers with data scarcity concerning sub-daily rainfall events. There-
fore the selection is very restrict to the existing (and operating) stations. The stations in Gilbués, Aiuaba and Sumé (Fig. 2) were maintained by research groups (Simplício et al., 2020; de Figueiredo et al., 2016; Srinivasan and Galvão, 2003) and only the station of Sobral is maintained by the Brazilian Water Management Agency (ANA). Those four stations presented consistent measurements over at the least two years without gaps. Another constrain for the selection of stations was the proximity to the sediment control equipment. Again, the stations in Gilbués, Aiuaba and Sumé were installed to monitor experimental basins
and are inside the catchment areas. The Sobral Station was chosen because it is in the Várzea da Volta catchment and is the closest to Acarape under the same climate conditions. For a detailed map of stations in the region please refer to the Supporting Materials.





**Table 2.** Study areas information. Lines with same colour indicate areas that share automatic rain gauge data.

| Basin | Area (km²) | Control system | Land use | Location | Catchment position Lon | Catchment position Lat | Bathymetry First[a] | Bathymetry Second | Time Series | AWS Name | AWS Position Lon | AWS Position Lat | Recording Start | Recording End |
|---|---|---|---|---|---|---|---|---|---|---|---|---|---|---|
| Canabrava | 2.9 | Reservoir | Agriculture and open range cattle rasing | Ceará | 39.56 W | 6.97 S | 1944 | 2000 | 57 | Aiuaba | 40.22 W | 6.69 S | 2004 | 2014 |
| Aiuaba | 11.53 | Reservoir | Conservation area with native vegetation (Caatinga) | Ceará | 40.24 W | 6.65 S | 2003 | 2009 | 7 | Aiuaba | 40.22 W | 6.69 S | 2004 | 2014 |
| Várzea da Volta | 155 | Reservoir | Agriculture and open range cattle raising | Ceará | 40.62 W | 3.50 S | 1917 | 1997 | 81 | Sobral | 40.36 W | 3.69 S | 2017 | 2019 |
| Acarape | 208 | Reservoir | Agriculture and open range cattle raising | Ceará | 38.69 W | 4.20 S | 1924 | 1999 | 74 | Sobral | 40.36 W | 3.69 S | 2017 | 2019 |
| Sumé 2 | 0.0107 | Sediment load | Experimental area - Preserved vegetation | Paraíba | 36.88 W | 7.67 S | | - | 10 | Sumé | 36.88 W | 7.67 S | 1982 | 1991 |
| Sumé 4 | 0.0048 | Sediment load | Experimental area - Degraded land without vegetation | Paraíba | 36.9 W | 7.66 S | | - | 10 | Sumé | 36.88 W | 7.67 S | 1982 | 1991 |
| Gilbués | 0.0004 | Check dam | Abandoned land under desertification process without vegetation | Piauí | 45.34 W | 9.88 S | 2018 | 2019 | 1 | Gilbués | 45.34 W | 9.88 S | 2018 | 2019 |

[a] The first bathymetry corresponds to the topography in the year of construction.





Experimental data was used to estimate sediment yield (de Araújo et al., 2006; Morris and Fan, 1998). We used bathymetric assessments from different years of the reservoirs of five catchments (Canabrava, Aiuaba, Várzea da Volta, Acarape and
Gilbués) to estimate the total siltation ($\Delta V$ – see Eq. 9). Direct data of sediment yield (SSY) was available at the micro basins in Sumé, where monitoring is carried out eventwise (Srinivasan and Galvão, 2003). Table 2 lists the type and timing of available sediment yield data. For each catchment we obtained the time series of daily rainfall from FUNCEME (2019). Sub-daily measurements are scarce and available for the whole study period only in one station in Gilbués (Simplício et al., 2020). and one in Aiuaba (de Figueiredo et al., 2016), the basins with the shortest and most recent time series. Assuming similar climatic
and environmental conditions, we used the data from the Aiuaba station for the analysis of Canabrava, and from Várzea da Volta for Acarape.

## 3 Results

### 3.1 Probability distributions functions - MEDRID

Table 3 presents the entropy-based calibrated parameters for the B3 (beta distribution), G2 (gamma distribution) and G3
(generalised gamma distribution). Those values were obtained by solving the systems of equations in Table 1. In Figure 3 we present the model evaluators of distributions at the four stations. From the method evaluators we can observe that B3 represents poorly the distribution when compared with the gamma distributions (Figure 3). The G3 performs slightly better than G2. From Table 3 we see that the parameter $c$ of the generalised gamma does not sufficiently approach the unit (when $c = 1$, the gamma and generalised gamma are equal). The strict two-parameter gamma distribution (G2) does not quite represent the process, but
less skewed function G3 does.

| | B3 | | G2 | | G3 | | |
| --- | --- | --- | --- | --- | --- | --- | --- |
| | $a$ | $b$ | $a$ | $b$ | $a$ | $b$ | $c$ |
| **Sobral** | 1.124 | 4.316 | 0.250 | 1.525 | 0.066 | 2.114 | 0.678 |
| **Aiuaba** | 1.584 | 10.686 | 0.138 | 1.855 | 0.004 | 3.306 | 0.488 |
| **Gilbués** | 0.696 | 2.691 | 0.777 | 0.953 | 0.390 | 2.099 | 0.812 |
| **Sumé** | 0.955 | 5.398 | 0.740 | 0.911 | 0.269 | 1.410 | 0.818 |

**Table 3.** Equations parameters for the $D/H$ distribution. $a$, $b$, and $c$ are the parameters as described in Table 1. The data used to calibrate the parameters are available in the supplementary material.

Two probability distribution functions were tested for the ratio $I_{30}/H$. The beta distribution (B3) and uniform distribution allow an explicit definition of lower and upper boundaries. For the Sobral, Aiuaba and Gilbués stations the uniform distribution presented a much better results, with Nash-Sutcliffe Efficiency (NSE) as high as 0.98, while the beta distribution had an efficiency lower than 0.50 (Figure 4). In the Sumé station both B3 and uniform distributions had similar performance with
NSE of 0.98 and 0.99 respectively. In this work we used the uniform distribution for the modelling in all regions.



Additionally, using statistical measures, we calculated the information loss resulting from using he PDF calibrated for one region into another (Eq. 2 and 3). We compared the four stations with sub-daily data among themselves. The measures (symmetric divergence and Kolmogorov-Smirnov distance) for the variable $D/H$ are given in Table 4.

| | Symmetric Divergence | | | | | Kolmogorov-Smirnov Distance | | | |
|---|---|---|---|---|---|---|---|---|---|
| | Sobral | Aiuaba | Gilbués | Sumé | | Sobral | Aiuaba | Gilbués | Sumé |
| Sobral | 0 | 0.396 | 2.419 | 0.194 | Sobral | 0 | 0.242 | 0.550 | 0.152 |
| Aiuaba | 0.396 | 0 | 4.987 | 1.073 | Aiuaba | 0.242 | 0 | 0.719 | 0.365 |
| Gilbués | 2.419 | 4.987 | 0 | 1.187 | Gilbués | 0.550 | 0.719 | 0 | 0.404 |
| Sumé | 0.194 | 1.073 | 1.187 | 0 | Sumé | 0.152 | 0.365 | 0.404 | 0 |

**Table 4.** Values of Symmetric Divergence and Kolmogorov-Smirnov Distance for the Generalised Gamma distribution of $D/H$. The higher the value, the greater the difference between the probability distributions.

These measures indicate that there is a considerable difference in the duration-precipitation ($D/H$) distribution in Gilbués over the other three regions.

Sobral and Sumé also appear to be very similar despite the distance between them. Located in the Brazilian Semiarid Region, the station in Sobral, Sumé and Aiuaba are under the same major atmospheric process for rainfall formation (the Inter-Tropical Convergence Zone - ITCZ) and have a similar rainfall regime (more than 70% of the annual precipitation concentrated in three months) and amount (500-600 mm yr[-1]). Gilbués has a higher precipitation (1200 mm yr[-1]) and better temporal distribution. Therefore, based on statistical distances (Table 4) and regional characteristics, Sobral and Sumé are most similar and have the lowest information loss when (quality) data from one station is used for the other region. Aiuaba is also similar to Sumé and (especially) to Sobral. Gilbués has particular PDF parameters, with both $D_{KL}$ and $\delta$ significantly higher when compared with the other three stations.



**Figure 3.** Probability distributions and the performance evaluators for the variable $D/H$.



**Figure 4.** Probability distributions and the performance evaluators for the variable $I_{30}/H$.





## 3.2 Sediment yield modelling

Two models were tested to assess sediment yield: a classic model consisting of the multiplication ULSE gross erosion ($\bar{\varepsilon}$) and empirically-based SDR (Maner, 1958), hereby called model M1; and the proposed MEDRID+SYPoME model (M2).

In Table 5 we present the output of the combination of MEDRID method and SYPoME model (M2) for the seven study areas. Average modelled sediment yield at the outlet varied between 5 (Aiuaba) and 2346 (Sumé 4) Mg km$^{-2}$ yr$^{-1}$ and SDR between 5.9% (Várzea da Volta) and 29.7% (Gilbués). The outputs for sediment yield and SDR of model M2 passed in the normality test

Shapiro and Wilk (1965) and we obtained the confidence interval (p = 0.01) using a Gaussian distribution. M1 is a deterministic model, thus has only one single output, presented in Figure 5.

| Basin | Sediment yield (Mg km$^{-2}$ yr$^{-1}$) | | | | SDR (%) | | | |
|---|---|---|---|---|---|---|---|---|
| | $\mu$ | $\sigma$ | C.V. | C.I | $\mu$ | $\sigma$ | C.V. | C.I |
| Canabrava | 664.5 | 24.9 | 4% | 12.5 | 13.9 | 0.2 | 1.4% | 1.04 |
| Aiuaba | 5.0 | 1.2 | 25% | 0.6 | 14.8 | 4.2 | 28.4% | 2.12 |
| Várzea da Volta | 418.2 | 20.2 | 5% | 10.1 | 5.9 | 0.4 | 7.3% | 0.22 |
| Acarape | 189.5 | 9.1 | 5% | 3.1 | 8.3 | 0.7 | 8.1% | 0.23 |
| Sumé 2 | 13.1 | 1.8 | 14% | 0.9 | 23.5 | 2.6 | 11.1% | 1.32 |
| Sumé 4 | 2345.6 | 264.1 | 11% | 132.9 | 20.4 | 3.0 | 14.6% | 1.50 |
| Gilbués | 2141.7 | 540.5 | 25% | 272.0 | 29.7 | 8.9 | 29.9% | 4.47 |

**Table 5.** Modelled values (M2) of sediment yield and SDR for the study areas. The values are shown in terms of average ($\mu$), standard deviation ($\sigma$) and coefficient of variation (C.V.). Confidence Intervals (C.I.) of the average calculated for p = 0.01.

In Figure 5 we present two plots. Figure 5a shows modelled (M1 and M2) and measured values of siltation rate (siltation rate per unit of area) and on Figure 5b the modelled (M1 and M2) values of SDR. The siltation rates generated by our approach (M2) clearly outperform those based on deterministic methods (M1). When assessing average sediment yield for each area, our

model also outperforms the deterministic model for all experimental basins, with an error reduction by a factor of at least 2 and as high as 20 (Table 6). Also, the new methodology (M2: MEDRID+SYPoME) presented better performance evaluators (NSE = 0.96 and RMSE = 608.6 ton km$^{-2}$ yr$^{-1}$) than the conventional (M1) approach (NSE = -4.49 and RMSE = 3286 Mg km$^{-2}$ yr$^{-1}$)

By comparing the values of siltation rate in Figure 5a with Land Use and Land Cover (Table 2) we can draw a strong correlation between them. Catchments with preserved vegetation, such as Aiuaba and Sumé 2, have the lowest siltation rate, over two order

of magnitude lower than degraded regions, such as Sumé 4 and Gilbués. Basins with the presence of agriculture (Canabrava, Várzea da Volta and Acarape) presented intermediary rates, although ten times larger than preserved regions.

Figure 5b shows the modelled average SDR (for the whole time series) of the basins obtained by M2 and M1 (Eq. 10). Considering the area of the basins (Tab. 2), we can observe a dependency of the SDR to the catchment area. Although M2 also showed similar tendency, its values of SDR are systematically lower than M1's. It is interesting to note that for the catchments

Canabrava, Acarape and Várzea da Volta there is almost no dispersion of SDR values. This is due to the long time series for





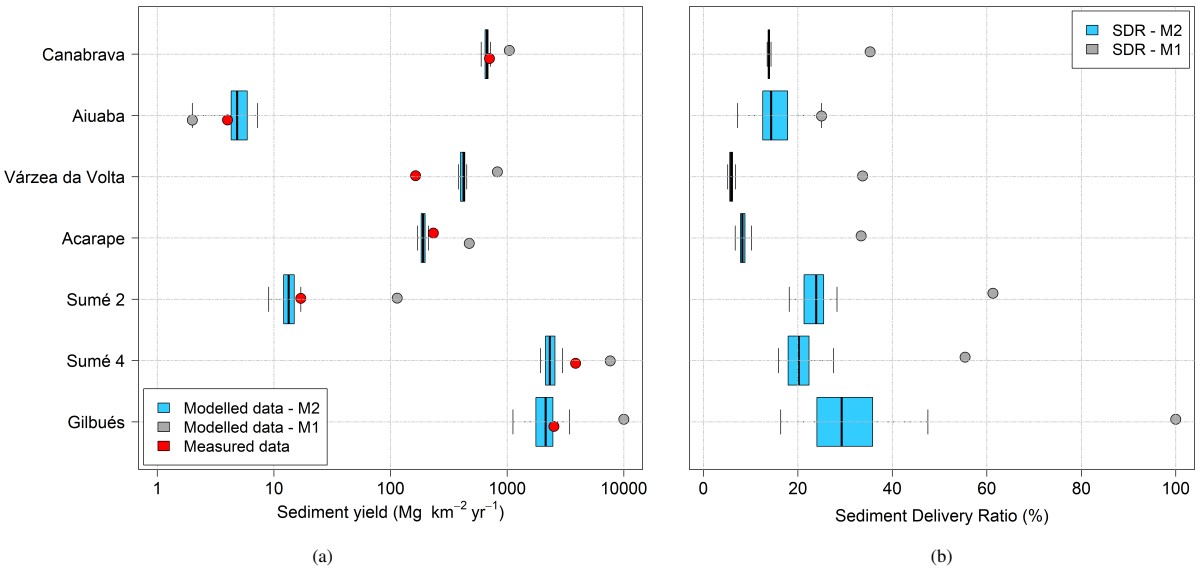

(a)                                                                                    (b)

**Figure 5.** M1 and M2 outputs of (a) sediment yield and (b) SDR. Red dots in (a) indicate the measured values of sediment yield.

those experimental areas. With a long temporal series, the averaging of the SDR of all events tends to a narrow range of values that can be understood as the basin SDR.

| Name | Brune Coefficient | Sediment yield (Mg km$^{-2}$ yr$^{-1}$) | | | Relative Error (%) | |
|---|---|---|---|---|---|---|
| | | Measured | Modelled M1 | Modelled M2 | Modelled M1 | Modelled M2 |
| Canabrava | 0.98 | 704 | 1042 | 664 | *48.0%* | *-5.6%* |
| Aiuaba | 1.00 | 4 | 2 | 5 | *-50.0%* | *27.5%* |
| Várzea da Volta | 0.95 | 164 | 824 | 418 | *402.3%* | *155.1%* |
| Acarape | 0.98 | 233 | 473 | 191 | *102.9%* | *-18.1%* |
| Sumé 2 | 1.00 | 17 | 114 | 13 | *570.6%* | *-21.8%* |
| Sumé 4 | 1.00 | 3857 | 7644 | 2314 | *98.2%* | *-40.0%* |
| Gilbués | 1.00 | 2518 | 10305 | 2142 | *309.3%* | *-14.9%* |
| **NSE** | | | **-4.49** | **0.96** | | |

**Table 6.** Measured and modelled values of siltation rate (Mg km$^{-2}$ yr$^{-1}$). M1 represents the classic model using empirically-based SDR (Manner, Eq. 10) and M2 the proposed MEDRID+SYPoME model.

## 4 Discussion

The complexity of hydrological processes can be better modelled with help of stochastic approaches (Sidorchuk, 2009; Singh,
2011). Sidorchuk (2009) proposed a path for sedimentological models relying on the combination of deterministic and probabilistic models in a so-called third generation erosion model, to which our method belongs. By introducing stochastic routines and calibrating parameters with the principle of maximum entropy, we extracted from the scarce data more valuable information





than by employing deterministic models, and even preserved the local characteristics of each region. The method performed well across a large range of time series and catchment-area scales.

## 4.1 Probability distributions functions - MEDRID

In literature (Singh, 1998; Bhunya et al., 2007; Brigandì and Aronica, 2019; Martinez-Villalobos and Neelin, 2019) many probability distribution functions are related to precipitation processes (e.g. Gamma, Power-Law, Exponential); especially concerning its duration (e.g. Gamma, Weibul, Lognormal). From Figure 3, we conclude that, although the Gamma distribution (G2) does reproduce the $D/H$ ratio, the Generalised Gamma distribution yields the best results in all study areas. Its better fit to the measured data appears to be related to the high complexity (uncertainty/entropy) involved on rainfall events, when many factors interact simultaneously. In such conditions, a less constrained distribution, as the G3 allows for more flexibility and calibration. With one additional parameter, the function becomes more adaptable to the peculiarities of each region in comparison with G2. This is confirmed by the values obtained for the parameter $c$, which never approximate to one (Tab. 3). Table 1 shows that a parameter c equal to one reduces a generalised gamma distribution to a conventional one (G2).

Information entropy is a measure of uncertainty (Jaynes, 1957). Therefore, the PoME delivers the probability distribution function that maximises the uncertainty under a set of constrains and avoids unproven assumptions (Chiu, 1991). It can be proven that the uniform distribution, as the one obtained for $\frac{I_{30}}{H}$, has the highest uncertainty (see Jaynes, 1957).

In the selection of the best distribution using the PoME, additionally to the constrains listed in Table 1, there is an implicit assumption taken: that the data follow a specified distribution (i.e. beta, gamma, uniform, etc.). Silva Filho et al. (2020) pointed out that the selected constrains of the PoME have to be relevant to the studied variable and that additional constrains do not necessarily lead to better results. Therefore, as we see in Figures 3 and 4, the narrowest distribution does not necessarily suit best the model. The constrains-quality trade-off problem becomes clear in the modelling of rain intensity (section 3.1), where the most suitable distribution is the uniform one. Such a result occurs because the unproven implicit constrain (the distribution itself) showed not to be valid.

The use of a uniform distribution for intensity implies that a stochastic approach is more valid than regression curves, as previously proposed by Avila and Avila (2015), Alencar et al. (2020), and Dash et al. (2019). Therefore, in stochastic models, a more realistic approach to be adopted is the uniform distribution, as expressed in Equation 11. The value of 30-minute intensity ($I_{30}$) can vary between 0 – in the case of $H \to 0$ – and 2H (for a precipitation with duration lower than 30 minutes). Equation 11 is a general equation and does not depend on calibration. Nevertheless, the implementation of Eq. 11 also requires a Monte Carlo approach, as presented in section 2.3, with draw of multiple random seeds ($\#_{rand}$).

$$I_{30} = \frac{H}{D} + H\left(2 - \frac{1}{D}\right)\#_{rand} \text{ such that } \frac{I_{30}}{H} \in (0, 2] \tag{11}$$





In terms of regionalisation of the MEDRID method, equations calibrated using data from a gauged catchment can be used in ungauged regions, provided that they have similar relief and climatic conditions, thus reducing the loss of information. It is important to note that geographic proximity between the station and the application site is not enough to guarantee better

parameter homogeneity and, thus, good model performance. The equations from Sumé and Sobral are remarkably similar, although they are more distant from each other than to Aiuaba. Nevertheless, the conditions of the Aiuaba catchment, which is higher and prone to orographic precipitation, may explain its distinction from the others. Finding the causes of similarities between areas, however, surpasses the scope of this work. Still, from analysis of relief and climate of the studied areas and based on the statistical distances(Table 4), we can build a map of possible factors that influence such similarity (Fig. 6). The

relative position of each area in Figure 6 is based on geographical location. The connecting lines indicate how similar the areas are to each other.

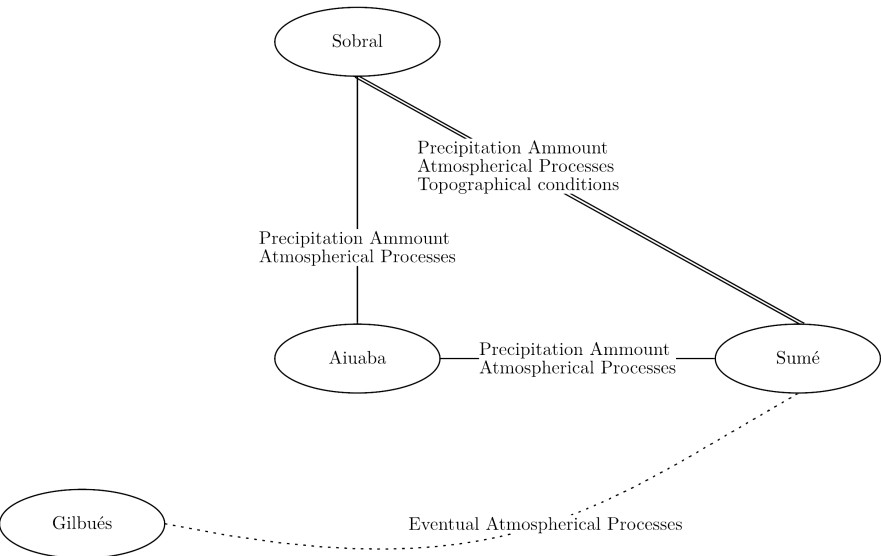

**Figure 6.** Clustering (connections) of regionalised PDFs and possible influencing factors for the similarities, based on relief and climate conditions. Note that nodes were positions to roughly match the geographical location of each study area (no scale – Fig. 2).

de Araújo's (2017, see section 2.1.1 of this paper) method of precipitation down-scaling, although simpler, has two problems. Firstly, each precipitation event is processed by the model only once, using an averaged duration as input. This reduces the freedom of the model to simulate extreme cases. The model by de Araújo (2017) also tends to represent the process by a linear

function, after the averaging (Figure 7). And secondly, the author's approach assumes a normal distribution of duration and daily precipitation. It is also assumed that both distributions are related by an unknown scaling factor $\chi$ (Eq. 5c). None of these assumptions could be confirmed by experimental data.

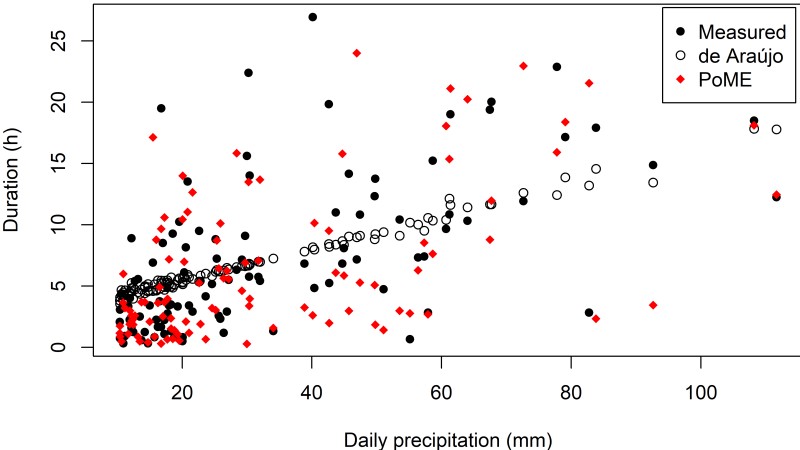

**Figure 7.** Scatter plot of daily precipitation and duration for Aiuaba. Note that both methods depend on random seeds, therefore the points position in the plot is not fixed, but rather an example. Other examples are available in the supplementary material.

## 4.2 Sediment yield modelling

In all cases the MEDRID+SYPoME model (M2) performed better than the deterministic model (M1) with empirically-based
SDR. As shown in Figure 5 and in Table 6, the relative error was reduced nine-fold, on average. Except for Várzea da Volta,
the average error was 21%, five times smaller than the average error for M1. Also excluding Várzea da Volta, the performance
of M1 was similar to values obtained from literature (see Risse et al., 1993). The Nash-Sutcliffe Efficiency of event-wise
sediment yield calculated for the catchments of Sumé 2 and 4 (0.52 and 0.47, respectively) can be classified as satisfactory
since its efficiency is marginally equal to 0.50 (Moriasi et al., 2007). These are, nonetheless, important results, especially
considering the little information required to achieve them. The efficiency of the model for total siltation rate is 0.96 (Table
6), its classification ranges from very good (Moriasi et al., 2007) to good (Ritter and Muñoz-Carpena, 2013). It supports the
argument that stationary parameters such as relief (in our temporal analysis scale) play a relevant role for sediment delivery
mechanisms (Simplício et al., 2020); they, therefore, increase the performance of the model over time.

Both models perform poorly in the assessment of siltation of the Várzea da Volta reservoir (see also Gaiser et al., 2003). This is
mainly caused by the peculiarity of its catchment topography and lithology. As illustrated in Figure 8, the upper (southern) part
of the watershed is formed by a plateau ending in a cliff of over 500 meters in depth formed by a soil that is prone to erosion
(USLE parameter $K = 0.032$ Mg h MJ$^{-1}$ mm$^{-1}$ – Gaiser et al., 2003). The lower portion of the watershed is mostly flat, and its
soil has a higher permeability, promoting an interruption of connectivity and therefore reducing the SDR, similar to the process
identified by Medeiros and de Araújo (2014) in a flat area upstream the Benguê Reservoir, North-eastern Brazil. Our model
(M2) was not able to describe such behaviour, although it significantly reduces the error when comparing to the conventional
methodology (M1).

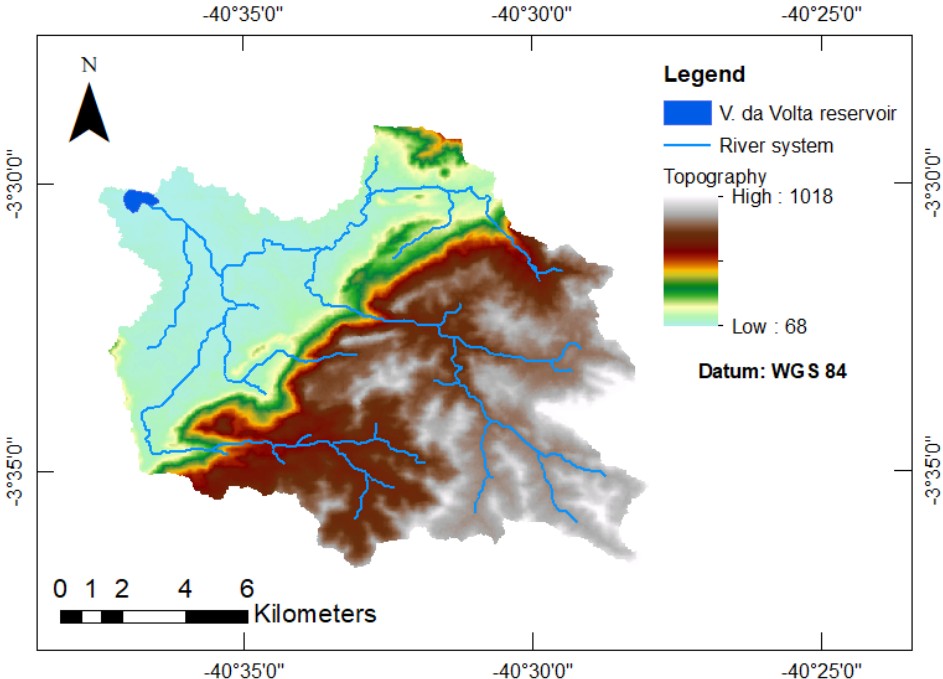

**Figure 8.** Topography and river system of the Várzea da Volta Catchment area.

One limitation of this study is the use of the Universal Soil Loss Equation to assess gross erosion. The USLE does not directly address gully erosion (Wischmeier and Smith, 1978). Nevertheless, gullies may be major sediment sources (Bennett and Wells, 2019), especially in degraded areas such as Sumé 4 and Gilbués (Srinivasan and Galvão, 2003; Simplício et al., 2020).

## 5   Conclusions

We propose a novel method to downscale duration and intensity of precipitation for erosion modelling based on daily data. The best probability distribution function for the duration-precipitation ratio ($D/H$) is the generalised gamma distribution (NSE = 0.98). For the ratio $I_{30}/H$, the uniform distribution (NSE = 0.47) performs best. The MEDRID method presents resilience to regionalisation, therefore demanding less climatological stations to cover a large area and allowing the implementation of the model in regions with data scarcity.

Using the downscaled duration and $I_{30}$ intensity generated by MEDRID, we are able to assess sediment yield with a higher accuracy than conventional USLE and relief-based SDR. The coupling MEDRID+SYPoME model allowed assessment of event-wise sediment yield and presented error six times smaller than the ones from conventional models. The new model (MEDRID+SYPoME), based on the combination of deterministic and entropy-based components improved substantially performance of assessment of sediment yield (NSE = 0.96) when compared with deterministic modelling.



Additional studies should be carried to test and assess the most suited probability distribution families to precipitation data, especially 30-minute intensity. Efforts are still necessary to validate the method's potential concerning regionalisation. It is not at all a trivial question, which factors (relief, climate, position, etc.) influence homogeneity between regions, and therefore produce similar PDF.

The MEDRID method can be used to assess rainfall sub-daily features (duration and 30-minute intensity). When coupled as MEDRID+SYPoME, the novel model provides accurate results of sediment yield across a wide range of catchment areas in catchments with areas of different orders of magnitude (from $10^{-3}$ to $10^{+2}$ km$^2$) and land use.

*Code and data availability.* Code and data are available at https://github.com/pedroalencar1/MEDRID-SyPOME.

*Author contributions.* Alencar worked on programming, data processing and analysis. Paton carried out data analysi. de Araújo contributed
with programming and data processing. de Araújo and Paton collaborated as supervisors of the work and of its conceptualisation. Alencar prepared the text with contributions of all authors.

*Competing interests.* The authors declare that they have no conflict of interest.

This work is part of the PhD work of **Pedro Alencar** and will be used in his dissertation.

*Acknowledgements.* This study was partly financed by the Coordenação de Aperfeiçoamento de Pessoal de Nível Superior – Brasil (CAPES;
finance code 001, CAPES/Print grant no 88881.311770/2018-01) and by Edital Universal (CNPq grant no. 407999/ 2016-7. Pedro Alencar is funded by the DAAD (award no. 91693642).





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
