# Peer review of "Maximum Entropy Distribution of Rainfall Intensity and Duration - MEDRID: a method for precipitation temporal downscaling for sediment delivery assessment."

_Hydrology and Earth System Sciences, 2021_

## Author Comment (AC1)

Hydrol. Earth Syst. Sci. Discuss., referee comment RC1
https://doi.org/10.5194/hess-2021-278-RC1, 2021 ©
Author(s) 2021. This work is distributed under the
Creative Commons Attribution 4.0 License.

[Figure]

**Author's reply on hess-2021-278**

Anonymous Referee #1
* * *
Referee comment on "Maximum Entropy Distribution of Rainfall Intensity and Duration – MEDRID: a method for precipitation temporal downscaling for sediment delivery assessment" by Pedro Henrique Lima Alencar et al., Hydrol. Earth Syst. Sci. Discuss., https://doi.org/10.5194/hess-2021-278-RC1, 2021
* * *
**Referee.** *The authors have reviewed the manuscript intitled "Maximum Entropy Distribution of Rainfall Intensity and Duration – MEDRID: a method for precipitation temporal downscaling for sediment delivery assessment" by Pedro Henrique Lima Alencar et al. In this manuscript, the authors presents a method (MEDRID) for precipitation temporal downscaling, then coupled with the SYPoME model to indirectly assess the MEDRID performance. Even though the results presented in the several catchments indicated that the MEDRID method have a good performance, but the authors should make it clear what the novelty of this study is. In addition, the description of how to couple with the SYPoME model is not clear, which makes a direct application difficult of this method. I would then suggest rejection of the article with invitation to resubmit. My major comments are described below.*

**Authors' response.** Dear Anonymous Referee #1, thank you very much for your comments, which we fully accept and comment in the following lines.

**Referee.** *(1) The novelty of the MEDRID method is not clear in this version. The Maximum Entropy Principle (MEP) has been widely used in many fields for the selection of an optimal distribution function. The authors seems just use the MEP to select an optimal distribution function for rainfall intensity and duration, if so, the article obviously lacks innovation. Thus, the authors should classify the novelty of the MEDRID method.*

**Authors' response.** We agree with the Referee that the mere selection of the optimal probability distribution function using the PME does not suffice for a paper to be accepted. However, the novelty of the manuscript goes beyond it. It lays in a robust method (based on many years of field data and on a physical and statistical / MEP approach) to adequately downscale daily rainfall data into sub-daily information that is useful to assess erosive processes. Thus, the use of the optimal distribution to assess sediment yield in data-poor areas is of relevance. Despite the novelty of the proposal and its good results, the authors agree that we failed making it clear in the manuscript. We also agree that MEDRID should be posed as a tool, not as the central message of the paper.

**Referee.** *(2) The description of how to calculate the series of the ratio D/H and I30/H is confused. Is D/H and I30/H relative to a rainfall event (may last for several days) or just relative to a daily rainfall (only one day)? Should all non-zero rainfall days be considered?*

**Authors' response.** We thank the Reviewer to address this issue and it will certainly be further explored in the revised text. The main rainfall formation process in the region is convective (and, less important, orographic), which causes intense precipitation events with limited duration. In fact, after 19 years of monitoring, we observe that less than 0.5% of the events last more than 24 h; and the longest event lasted 26 h. Because of this feature, and for simplicity's sake, we assume that events longer than 24 h are considered to occur on a single day.

**Referee.** *(3) The application of the D/H distribution, coupled with the SYPoME model, were described in Sect. 2.3. But I did not see relative description of how to use the $I_{30}$/H distribution for SYPoME model. The reliability should be improved.*

**Authors' response.** The authors agree with the Referee that this subject must be better explained in the revised text. The ratio ($I_{30}$/H) is used to assess the rainfall erodibility. This issue had been addressed in the supplements, but it can be brought to the main text, so as to implement the necessary clarification.

**Referee.** *(4) The authors indirectly assess the performance of the MEDRID method by comparing the M1 and M2 model. However, it can be found from the Figure 5 that the M2 error is systematically large, why? Does this affect the reliability of the comparison result?*

**Authors' response.** We thank the Referee for the comment; however, we believe that there is a misunderstanding at this point. In Figure 5 the red dots indicate the measured data. The model M2 output is indicated as a boxplot because the model is non-deterministic, and its output is a set of possible answers. In Figure 5a, we observe that in all (seven) catchments, the measured data is actually closer to the M2 output set than to the grey dots, which indicate the deterministic approach.

[Figure]

**Figure 5.** M1 and M2 outputs of (a) sediment yield and (b) SDR. Red dots in (a) indicate the measured values of sediment yield.

---

## Author Comment (AC2)

Hydrol. Earth Syst. Sci. Discuss., referee comment
RC2 https://doi.org/10.5194/hess-2021-278-RC2,
2021 © Author(s) 2021. This work is distributed
under the Creative Commons Attribution 4.0
License.

[Figure]

**Author's reply on hess-2021-278**

Anonymous Referee #2
* * *
Referee comment on "Maximum Entropy Distribution of Rainfall Intensity and Duration – MEDRID: a method for precipitation temporal downscaling for sediment delivery assessment" by Pedro Henrique Lima Alencar et al., Hydrol. Earth Syst. Sci. Discuss., https://doi.org/10.5194/hess-2021-278-RC2, 2021
* * *
Dear Anonymous Referee #2, we thank you very much for your comments on our work; we fully accept them.

**Referee.** *This manuscript by Alencar et al. addresses a very important topic to estimate the sediment yield in data scarce region. Although research topic and Introduction motivated me to look forward to further sections on Methodology and Results in the manuscript, I must say that it was very confusing from Section 2 onwards to follow because Authors kept referring to previous work by one of the co-authors de Araujo 2007 and 2017. I looked into supplementary material which was not really helpful. I suggest Authors to present the methodology clearly and as far as possible, independent from previous work. Otherwise, novelty of the approach becomes questionable.*

**Author's response.** The Methodology and results are being reshaped to increase clarity. The works from de Araújo (2007 and 2017) are referenced because our submitted paper is an advancement upon the two previous works. The first (de Araújo, 2007) presents the SYPoME formulation that allows sediment yield assessment. The second (de Araújo, 2017) presents an initial attempt to use daily data to downscale daily precipitation in ungauged catchments (same goal as our more recent work, however they did not use entropy).

**Referee.** *As a hydrologist, I was hoping to see the results of temporal downscaling in terms of a time series showing daily and subdaily information at the selected sites. There are many literature on the topic of downscaling daily to subdaily data, especially rainfall, but Authors actually went quickly into MEDRID-SYPoME description, overlooking the fact that the downscaled results need to be evaluated carefully as it is one of the major inputs for producing sediment yield.*

**Author's response.** Our focus with the MEDRID is to obtain duration and 30-minute intensity, necessary as input to the SYPoME model. As pointed out by Anonymous Referee #1, presenting the MEDRID in such centrality was a poor choice that is being corrected.

**Referee.** *I wonder the approch presented in this manuscript is area specific. The claim that the presented method is more accurate than the conventional method can't be justfied unless the approach is applied to different topographical and climatic regions. To set up a hydrological model to derive sediment yield at the outlet, one has to calibrate so many parameters using the observation data, I am not clear how all those complicated steps can be bypassed by using a set of equations.*

**Author's response.** The SYPoME model has been implemented both in multiple catchments Brazil and Switzerland with success and requires the calibration of parameters, which are not area specific. The downscaling method presented requires the definition (calibration) of the suited probability distribution function (a generalized gamma distribution). Finally, to validate the model we used measures of siltation that are hard and costly to obtain, particularly with the level of quality presented in this study, therefore only a limited number of study areas as available.

**Referee.** *The main results for sediment yield is presented in Figure 5. it would be better if the results are presented in terms of time series, not just one red dot showing total yearly sediment yield. Also 5% and 95% confidence limit on the modeled result can be shown in a time series plot.*

**Author's response.** This illustration can be prepared, however measured data are not available as time series, as presented in Table 2.

**Referee.** *Overall, Authors should very carefully highlight what is there in this work which is different from their previously published work. Present the methodology clearly so that readers don't need to refer to several other papers to understand it. Also, the results section should be made clearer with more plots.*

**Author's response.** Once again, we thank the Referee for their thoughtful comments, which will be taken into consideration and accepted. We are already working on the necessary improvements.